# Digital Twin for Analog Mars Missions: Investigating Local Positioning Alternatives for GNSS-Denied Environments

**DOI:** 10.3390/s25154615

**Published:** 2025-07-25

**Authors:** Benjamin Reimeir, Amelie Leininger, Raimund Edlinger, Andreas Nüchter, Gernot Grömer

**Affiliations:** 1Institute of Mechatronics, University of Innsbruck, 6020 Innsbruck, Austria; 2Department of Sport Science, University of Innsbruck, 6020 Innsbruck, Austria; 3Ludwig-Maximilians-University, 80539 Munich, Germany; a.leininger@campus.lmu.de; 4Department of Smart Automation and Robotics, University of Applied Sciences Upper Austria Campus Wels, 4600 Wels, Austria; 5Department of Computer Science XVII: Robotics, Julius-Maximilians-University, 97074 Würzburg, Germany; andreas.nuechter@uni-wuerzburg.de; 6Austrian Space Forum, 6020 Innsbruck, Austria; gernot.groemer@oewf.org

**Keywords:** analog space mission, human–robot interaction, AMADEE-24, mobile robots, 3D motion capture, digital twin, sensor data fusion, planetary exploration, Mars analog mission planning, situational awareness

## Abstract

Future planetary exploration missions will rely heavily on efficient human–robot interaction to ensure astronaut safety and maximize scientific return. In this context, digital twins offer a promising tool for planning, simulating, and optimizing extravehicular activities. This study presents the development and evaluation of a digital twin for the AMADEE-24 analog Mars mission, organized by the Austrian Space Forum and conducted in Armenia in March 2024. Alternative local positioning methods were evaluated to enhance the system’s utility in Global Navigation Satellite System (GNSS)-denied environments. The digital twin integrates telemetry from the Aouda space suit simulators, inertial measurement unit motion capture (IMU-MoCap), and sensor data from the Intuitive Rover Operation and Collecting Samples (iROCS) rover. All nine experiment runs were reconstructed successfully by the developed digital twin. A comparative analysis of localization methods found that Simultaneous Localization and Mapping (SLAM)-based rover positioning and IMU-MoCap localization of the astronaut matched Global Positioning System (GPS) performance. Adaptive Cluster Detection showed significantly higher deviations compared to the previous GNSS alternatives. However, the IMU-MoCap method was limited by discontinuous segment-wise measurements, which required intermittent GPS recalibration. Despite these limitations, the results highlight the potential of alternative localization techniques for digital twin integration.

## 1. Introduction

Future space missions beyond Earth’s orbit present unique challenges and require intensive human–robot interaction (HRI) for operational success [1]. Despite advances in autonomy since the Apollo era, humans would still outperform robotic systems in scientific exploration, particularly in anomaly detection, real-time data interpretation, and adaptive problem-solving [2]. NASA’s Mars rovers have demonstrated the potential of teleoperated and autonomous robotic exploration [3]. However, the constraints of long communication latencies and intermittent connectivity between Earth and Mars impose operational limitations. Hence, modern robots are best positioned to complement rather than replace human capabilities by eliciting intuitive interaction and a balanced human–robot partnerships to reduce astronauts’ workload and support scientific operations [1,4]. NASA continues to explore innovative HRI frameworks to enhance mission operations [3].

Planning and coordinating extravehicular activities (EVA) for planetary exploration beyond Earth’s orbit—especially those involving multiple actors such as astronauts, support rovers, and autonomous systems—is a complex task. A major challenge lies in the limited situational awareness available to Flight Planners on Earth due to communication delays and reduced inofrmation availability [5]. Recent advancements suggest that digital twin technology could help bridge this gap by providing a dynamic, real-time representation of the mission environment and system states [6,7]. However, for a digital twin to function effectively, accurate localization of human and robotic agents is essential to ensure safe navigation and reliable operational insights [8]. Since Moon and Mars are currently GNSS-denied environments—unless supported by significant satellite infrastructure—alternative localization methods are required. While various techniques have shown promise in controlled, isolated experiments, they have rarely been integrated or validated in the context of full-scale mission simulations [9,10,11]. Moreover, existing digital twin systems for planetary exploration remain largely conceptual and lack comprehensive testing in realistic, high-fidelity mission scenarios. Our real-world experience of human planetary exploration is limited, mainly relying on the Apollo missions, which highlights the need for high-fidelity simulations in analog environments on Earth [12]. Analog missions provide essential insights into the effectiveness of exploration by testing new methods and technologies in controlled, realistic conditions, helping to optimize EVA planning with a focus on time and risk management [13]. The Austrian Space Forum’s (OeWF) AMADEE-24 mission is designed to simulate Mars exploration, involving analog astronauts, mobile robotic platforms, and remote science teams. One of the mission’s key challenges is enabling intuitive and efficient interaction between astronauts and robotic assistants.

This study aims to present a comprehensive human–robotic digital twin developed for the AMADEE-24 analog Mars mission, evaluate its validity through experimental results, and explore its potential for future planetary exploration. Furthermore, by testing alternative local positioning methods, we aim to enhance the approach for use in planetary surface missions that do not rely on satellite-based geolocation systems (GNSSs).

## 2. Related Work

### 2.1. Digital Twin Technology in Space Exploration

Key factors in EVA planning include efficiency in traversing terrain while mitigating environmental hazards like steep slopes, difficult surfaces, and physical exhaustion. Remote sensing from satellites and drones can assess the accessibility of points of interest (POIs), revealing areas on the Moon and Mars with geological structures inaccessible by foot or rover due to topological and environmental hazards [14]. Digital elevation models, generated from aerial imaging, are also common in EVA planning for analog missions [5]. However, recent advancements have fostered the integration of more information. Including various data sources, such as pressure suits, life support systems, EVA tools, and spacecraft hardware, will become essential for effective EVA planning [15]. Recent path optimization algorithm developments based on hazard maps and metabolic workload models have enabled more sophisticated EVA planning tools [16]. While these technical developments largely rely on static models of operators and the environment, integrating real-time data and advanced simulation models marks the next evolution. A digital twin goes beyond traditional static models by dynamically replicating the characteristics and behaviors of its physical counterpart. Digital twins in the realm of space enigineering are already used in spacecraft lifecycle management [17] or health and usage monitoring of exploration rovers [18]. But this technology also allows for real-time monitoring and predictive simulations, capturing the evolving conditions of astronauts, collaborative rovers, and the exploration environment [6]. The shift from static to dynamic modeling provides a more precise and responsive approach to exploration in dynamic, changing environments, with the potential to enhance the health and performance of astronauts [7]. Reimeir et al. proposed a framework for creating a digital twin for planetary explorations [19]. The framework aims at integrating data sources along the methodological pathway of the exploration cascade, from remote sensing data (e.g., satellites, drones) over teleoperated rover explorations to high-fidelity data from human operators during EVA [20]. The digital twin is composed of two fundamental model concepts—the environmental model and one or more operator models. Precise localization of each operator in the environmental model is an essential part in this approach, particularly for movement analysis and biomechanics of the astronauts and the traverse performance of the rovers, which are highly dependent on local surroundings and surface conditions.

### 2.2. Localization of Astronauts and Rovers

Alternative methods to GNSS-based navigation are developed and tested due to their need in GNSS-denied environments like Moon and Mars. Among the different methods, IMU-based navigation and optical navigation with Light Detection and Ranging (LiDAR) sensors are amongst the most promising approaches.

Researchers conducted experiments in a desert analog environment using a single foot-mounted IMU. Their short rectangular traverse (107 m) resulted in an average localization error of 1.2 m (~1%) and a relative closure error of 10.2% [9]. However, error increased with distance, reaching 134.1 m (~3%) for a 4.8 km traverse and a closure error of 24.4%. Studies conducted in controlled environments, such as athletics tracks or around campus laboratories, demonstrated improved accuracy by integrating multiple IMUs. Reported errors were ranging from 0.5% to 2% across walking distances of 80 to 400 m [21,22]. Most studies employed either single or multiple IMUs, but did not foster an approach where IMU data was integrated with a kinematic model to track the 3D walking movement of the participants. Additionally, experiments were conducted with simplified traverse paths (e.g., straight lines or loops) and did not simulate EVA conditions with realistic movement patterns. However, 3D motion capture (MoCap) already plays a crucial role in analyzing and optimizing an astronaut’s range of motion in a space suit [23]. These studies are mainly conducted using optical MoCap systems in controlled laboratory settings such as NASA’s Neutral Buoyancy Laboratory, which are constrained to the field of view of the calibrated stationary cameras [23,24]. In response to the limitations of optical methods, inertial measurement unit-based motion capture (IMU-MoCap) can serve as a promising alternative for tracking astronaut movements in space suits during EVA [25,26]. Center of mass (CoM) trajectories captured by IMU-based MoCap during gait analysis proved accuracy comparable to optical systems, with average deviations of only 3.6 mm [27]. However, this analysis only focused on a few steps and did not localize individuals over longer durations, like during EVAs. In contrast to optical systems, movements are recorded in a local coordinate system, and accurate locomotion of the whole body relies heavily on the accurate representation of the human kinematic model, where the IMU data is projected on. Ziegler et al. utilized the Xsens MVN 3D MoCap system with the full-body kinematic model to track walking loops on a university campus (300–350 m), including rapid movements and small turns [10]. Reported localization errors at fixed checkpoints ranged from 2 to 8 m. However, as with other IMU-based systems, translational drift accumulated over time, leading to closure errors exceeding 15 m over 10 repeated loops.

A different localization approach, often tested in robot navigation, leverages LiDAR sensors. A critical aspect of ensuring safe and efficient robotic navigation in unknown extraterrestrial terrains is the accurate and robust use of the SLAM algorithm. The original LiDAR Odometry And Mapping (LOAM) SLAM algorithm estimates motion by extracting edge and planar features from LiDAR scans and minimizes the distance between these features in successive frames. This allows accurate ego-motion estimation even in sparse or repetitive environments. Lightweight and Ground-Optimized LiDAR Odometry And Mapping Bundle Optimization Refinement (LeGO-LOAM-BOR) is a modular extension of the popular LeGO-LOAM SLAM system [11]. It offers enhanced robustness in rough terrain, reduced drift through global pose graph optimization, and efficient integration with multiple sensor modalities. In our approach, we rely on the LeGO-LOAM-BOR SLAM method, which has proven effective for real-time pose estimation and map generation in various robotic applications. A study by Shan et al. [11] demonstrated the use of LeGO-LOAM-BOR in a structured indoor environment and reported average trajectory deviations below 0.15 m compared to ground truth, validating its accuracy and robustness.

Apart from rover self-localization, LiDAR sensors are widely used for human detection in robotics, autonomous driving, and surveillance due to their capability to provide high-resolution 3D point clouds. Traditional methods rely on geometric segmentation, feature extraction, and classification techniques [28] or machine learning algorithms [29] to distinguish humans from other objects. Several studies have demonstrated the effectiveness of LiDAR-based human detection. One approach uses histogram-based feature descriptors for point cloud segmentation to identify pedestrians [30]. Another integrates LiDAR and camera data within a learning-based framework to enhance classification accuracy [31]. More advanced solutions employ end-to-end deep learning architectures that directly process 3D LiDAR scans, achieving higher performance than traditional clustering techniques [32]. Adaptive Clustering approaches have been developed in dynamic, unstructured environments to handle variations in point density, occlusions, and sensor noise. A dynamic distance threshold was introduced to enhance detection rates in crowded scenes [33]. Additionally, adjusted segmentation parameters in response to changing environmental conditions increase robustness and adaptability of the clustering process [34].

## 3. Methodology

### 3.1. Human–Robotic Analog Mars Mission AMADEE-24

The data acqusition to develop the digital twin was embedded in the comprehensive, high-fidelity space mission simulation AMADEE-24. The AMADEE analog research program, led by the Austrian Space Forum, simulates human–robotic Mars exploration in terrestrial environments. The AMADEE missions provide a controlled environment for refining scientific methodologies, deploying technology, and developing operational strategies for future Mars exploration. The primary focus of each mission is the execution of scientific experiments, which occur within the habitat or in the surrounding terrain. Experiments conducted outside the habitat are performed by remotely controlled robotic systems or through EVAs carried out by analog astronauts. During EVAs, two astronauts wear the Aouda space suit simulator while two crew members inside the habitat oversee and support the operations via radio communication. Communication with the Mission Support Center (MSC) on “Earth” is time-delayed by ten minutes and is limited to text-based messaging. An onsite support team ensures logistical organization and safety but remains invisible to the analog astronauts to preserve simulation fidelity. The Aouda space suit simulator, available in two versions (AoudaX and AoudaS), replicates the constraints and procedures of real space suits for planetary surface exploration. Equipped with integrated sensors, the suits monitor and transmit the astronauts’ vital signs via a dedicated Wi-Fi network with directional antennas, enabling real-time telemetry to the habitat crew.

The AMADEE-24 mission was conducted in the Ararat region of Armenia from 7 March to 5 April 2024. After a bridgehead phase, the isolation period began on 15 March but was interrupted on 19 March due to technical issues and adverse weather conditions, lasting three days. Isolation resumed on 22 March. During this phase, six analog astronauts (two female, four male) were isolated in a dedicated habitat and conducted various robotic, psychological, and life science experiments. The EVA experiments conducted in this study were performed according to the Declaration of Helsinki and approved by the Board for Ethical Questions in Science of the University of Innsbruck (Approval Number: 23/2024).

### 3.2. Study Design: Digital Twin Integration and Operator Localization

The digital twin developed for the AMADEE-24 analog mission is building upon the methodological framework proposed by Reimeir et al. [19] (Figure 1). The environmental model, which includes the habitat, surrounding topography, and ambient weather conditions like air temperature and solar radiation, is characterized by relatively slow system dynamics. The digital elevation model (DEM) generated from drone scans served as the base information for the environmental model and allowed to generate an elevation profile visualized in 3D using Blender 4.2. Additionally, aerial images from the test site were applied as textures to enhance visual appearance of the model. The habitat was constructed after the initial drone scans for the DEM were recorded. Therefore, it was subsequently modeled based on its architectural floor plans and positioned in the model using the World Geodetic System 1984 (WGS84) coordinates of the habitat’s cornerstones. Each operator model is defined by two main aspects: its position within the environmental model at a specific point in time, and its modeled behavior, which reflects either the technical characteristics of a rover or the physiological and biomechanical traits of a human operator. Global Positioning System (GPS) served as the default navigation system during the AMADEE-24 mission. Our experiment leveraged two alternative local positioning methods: IMU-based MoCap for human operators and LeGO-LOAM-BOR SLAM algorithm and Adaptive Cluster Detection based on LiDAR scans for rover and astronaut localization, as proposed by Edlinger et al. [35,36].

The experiment procedure took between 30 to 40 min and foresaw four different geological sample tasks at distinct POIs, approximately 30 to 50 m apart. The POI varied for each EVA. The nominal protocol envisioned the execution and recording of the described procedure twice during an EVA, once at the beginning and once at the end of the EVA with other scientific experiments scheduled in between. Four EVAs were conducted where the experiment was performed nominally and one EVA had to be aborted due to a loss-of-signal procedure after the first experiment run. In total, nine recorded traverses are analyzed in this study.

To replicate rover and astronaut locomotion in the digital twin, a custom Blender add-on was developed, converting WGS84 coordinates and generating a continuous path mapped onto the modeled landscape (Figure 1). Due to relatively short distances covered during EVAs, the transformation of WGS84 GPS telemetry data to Cartesian coordinates of the environmental model was conducted using a simple equirectangular projection. The error introduced by the transformation is between 1.5 and 2.5‰, depending on the cardinal direction. To enable astronaut localization using IMU-MoCap, recorded whole-body movements were mapped onto an astronaut contour model. This approach enables the digital twin reconstruction of traverse locomotion and joint kinematics of the astronaut as they originally occurred. The local path generated through the rover’s LeGO-LOAM-BOR SLAM algorithm was initially set and calibrated using the rover’s GPS position.

Furthermore, on top of the SLAM localization of the rover, an Adaptive Cluster-based obstacle detection mechanism was applied based on generated point cloud. This process segments the environment into spatial clusters using Euclidean distance-based clustering, and each cluster is subsequently analyzed for its geometric properties. Combined with the LOAM-derived position, this allows us to track static and dynamic obstacles in 3D space. The alternative local positioning approaches for the operator models were subsequently compared to the default GPS positioning.

### 3.3. Data Acquisition

#### 3.3.1. Drone Scans and Aerial Imaging

High-resolution drone laser scans and aerial images were taken using a DJI Matrice 30T drone (SZ DJI Technology Co., Shenzhen, China) with a Zenmuse H20T camera (SZ DJI Technology Co., Shenzhen, China) before the mission as part of a 2-week measurement campaign. The area surveyed ranged from 250 to 500 m around the habitat, depending on accessibility and regions of interest. The covered area was about 125,000 m^2^, and over 1000 images were taken from three flights. Each time, the gimbal angle was adjusted (30°,60°,90°), and the images were taken with a grid distance of 2 m from a height of 150 m. The camera has a resolution of 20 MP, which, in scale, corresponds to approximately 5 cm per pixel. A digital elevation model was generated from the drone scans, with a resolution of 8000 × 6000 pixels and horizontal and vertical accuracies of 2 cm and 3 cm, respectively, using Pix4D software (4.8.2).

#### 3.3.2. iROCS Rover

The RTE rover (Rosenbauer Technical Equipment, Leonding, Austria) is a wireless, remote-controlled, electrically driven crawler-type vehicle measuring 800 mm in width, 1200 mm in length (pallet-sized), and weighing 375 kg, with a maximum payload of 625 kg. The iROCS rover uses a VK-162 G-Mouse USB GPS module (u-blox AG, Thalwill, Switzerland) with Circular Error Probable (CEP) of 2.5 m under good conditions (open sky, no obstructions). The GPS sample rate is 1 Hz. iROCS GPS was stored locally and transmitted after each EVA according to the experimental data transmission protocol. For AMADEE-24, we equipped it with a payload module featuring a 3D Velodyne VLP-16 LiDAR (Velodyne Lidar Inc., San Jose, CA, USA), an IMU for SLAM applications, geological tools, and sample collection boxes. At the front of the rover are two cameras designed to capture critical data during the mission. One of these is an analog camera that provides a real-time live stream for continuous monitoring. The second camera is a stereo camera with a Vertical Field of View of 41° and a Horizontal Field of View of 67°. Due to its high power capacity, the rover plays a pivotal role in supporting EVAs by serving as a mobile power source for the hardware of other experiments. In planetary exploration and space analog missions, autonomous robots support astronauts with navigation, transportation, and scientific tasks. In the AMADEE-24 mission, the interaction between astronauts and the rover is carefully coordinated to ensure safety, efficient sampling, and accurate data collection. The analog astronaut responsible for geological sampling (AA(S)) should walk in front of the rover at least 3 m, allowing the rover to operate while maintaining a safe distance from the astronaut (Figure 2). This positioning also helps ensure that AA(S) remains visible to the on-board cameras, critical for tracking and data analysis. The analog astronaut controlling the rover (AA(R)) should walk behind it, maintaining a clear line of sight of its locomotion. This enables AA(R) to effectively monitor and control the rover’s movement while maintaining high situational awareness. It also provides AA(R) with the best vantage point to observe the rover’s performance and adjust as needed.

#### 3.3.3. Aouda Space Suit Simulators

The two Aouda space suit simulators were developed to replicate operational and environmental constraints of a space suit during planetary surface exploration, particularly concerning systems integration, telemetry, and data acquisition [37]. Each suit has a total mass of approximately 45 kg, approximating the effective lunar EVA suit weight (96.2 kg) under Martian gravity conditions (0.38 g) [38].

While the suit includes biomechanical features to simulate limited mobility, its primary technical capability lies in the embedded sensor network and On-Board Data Handling (OBDH) system. It supports continuous monitoring and recording of environmental and physiological data. The Aouda suits utilize high-power GPS receivers, specifically the Navilock NL-604P (Tragant Handels- und Beteiligungs GmbH, Berlin, Germany) in AoudaX and NL-404P in AoudaS, with a CEP of 2.5 m. Aouda’s position was sampled every 5 s. Linear interpolation was used to upsample Aouda positions to 1 Hz to match iROCS sample rate. GPS was recorded in the WGS84 convention. Aouda’s network antenna enabled an approximate telemetry reach of up to 1 km, provided directional antennas were positioned optimally and a clear line of sight was maintained. Telemetry data includes inputs from multiple sensors, such as a three-lead Electrocardiogram (ECG) system to monitor cardiac activity, a CO_2_ sensor positioned at the rear of the helmet for respiratory gas analysis, and temperature and humidity sensors mounted on the left inside of the helmet. Additionally, all sensor data are continuously logged during EVAs and stored locally.

#### 3.3.4. Xsens Awinda

The 3D motion capture was conducted using the Xsens Awinda (Movella™, Enschede, The Netherlands) system on AA(S), while AA(R) controlled the iROCS rover [39]. Seventeen inertial measurement unit (IMU) sensors were mounted on distinct body segments of the upper and lower body according to the Xsens guidelines during suit donning, with some sensors (pelvis, chest, shoulders) integrated into the carrier system of the suit. Prior to the mission, anthropometric measurements were taken of each participant to scale the kinematic model to the respective participant, ensuring accurate motion capture. At the start of each EVA, a walking calibration trial was performed to initialize the system. The data receiver and local processing hardware were carried by the iROCS rover, while a crew member in the habitat remotely controlled the measurement system via network connection, managing data recording.

The motion capture system established a local coordinate system, independent of GPS orientation, and recorded the 3D rotations of body segments as well as the locomotion of the full-body model within that frame of reference. Data was sampled at 60 Hz, stored locally, and transferred to the MSC after each EVA.

### 3.4. Data Analysis: Alternative Astronaut and Rover Localization

#### 3.4.1. iROCS Localization Using LeGO-LOAM-BOR

LeGO-LOAM performs SLAM by extracting geometric features from incoming LiDAR scans and using these features to estimate the robot’s motion over time. Each LiDAR scan is divided into multiple scan lines (depending on the number of vertical channels of the sensor, such as 16 for a VLP-16), and individual points are analyzed for curvature. Based on this analysis, points are classified into edge features (sharp corners), planar features (flat surfaces), and less informative points, which are typically discarded or used for visualization only. The algorithm selects a subset of sharp edge points and flat surface points as key features for motion estimation. LeGO-LOAM estimates motion by aligning each LiDAR scan to a local map using edge and planar features. It minimizes point-to-line and point-to-plane distances via iterative optimization—typically using Iterative Closest Point (ICP) variants or nonlinear least-squares—to produce high-frequency, robust pose estimates. The back-end maintains a sliding-window local map, downsampled with a voxel grid filter for efficiency. This scan-to-map registration enables real-time, accurate trajectory tracking. In our setup, we used a 0.2 m voxel filter leaf size, a 2–60 m feature extraction range, and a Microstrain 3DM-GX5-IMU (Hottinger Brüel & Kjær GmbH, Darmstadt, Germany) with an update rate of 200 Hz.

We used the BOR (Bundle Optimization Refinement) extension to enhance LeGO-LOAM with global pose graph optimization. BOR detects loop closures by aligning the current scan with prior map segments and evaluating the fitness score (threshold: 0.15). When a valid loop is found, it adds a constraint to the pose graph. Every 10 s, or upon new loop detection, BOR refines all past poses via bundle adjustment to reduce accumulated odometric drift. Together, LeGO-LOAM and BOR form a hybrid SLAM solution that combines the responsiveness of real-time LiDAR odometry with the global consistency of pose graph optimization. This makes it particularly well-suited for applications requiring accurate mapping in large or complex environments, such as mobile robotics, planetary exploration, or search-and-rescue missions (Figure 2).

#### 3.4.2. Human Detection with Adaptive Clustering

Efficient and accurate preprocessing of LiDAR data is critical for reliable perception and mapping in autonomous robotic systems. A comprehensive preprocessing pipeline was used to enhance raw Velodyne VLP-16 LiDAR point clouds through three key steps: ground removal using a Random Sample Consensus (RANSAC)-based plane fitting method, statistical outlier removal to filter noise and spurious returns, and coordinate transformation to align sensor data with the robot’s reference frame or global map. The ground removal module robustly segments flat terrain from obstacles, significantly improving the quality of navigable space analysis. Outlier removal increases the density and consistency of the object point cloud, facilitating downstream tasks such as SLAM and object detection. Finally, coordinate transformation ensures seamless integration of sensor data within a multi-sensor fusion framework. The general objective is to minimize the error in position estimation by forming clusters of nearby position data points and refining the centroid of each cluster (Figure 2). The Adaptive Clustering algorithm used for astronaut localization operates in the following steps:1.**Data Acquisition:** The sensor network collects position data points at discrete time intervals.2.**Cluster Formation:** Data points are grouped based on their proximity in space. The distance between points is computed as(1)$$d\left(p_{i},p_{j}\right)=\sqrt{{\left(x_{i}-x_{j}\right)}^{2}+{\left(y_{i}-y_{j}\right)}^{2}+{\left(z_{i}-z_{j}\right)}^{2}}$$
where $d(p_{i},p_{j})$ is the Euclidean distance between two data points $p_{i}$ and $p_{j}$.3.**Adaptive Thresholding:** The algorithm adapts the clustering radius $r_{\max}$ based on the distribution of the points. If the distance between points exceeds a threshold, they belong to different clusters. This is given by(2)$$r_{\max}=\max\left(\frac{1}{n}\sum_{i=1}^{n}d\left(p_{i},p_{i+1}\right),\;\sigma \right)$$
where *n* is the number of points, and σ is a scaling factor.4.**Cluster Centroid Calculation:** The centroid of each cluster $C_{k}$ is computed as the average of the positions in the cluster:(3)$$C_{k}=\frac{1}{|S_{k}|}\sum_{p_{i}\in S_{k}}p_{i}$$
where $S_{k}$ is the set of points in cluster *k*, and $|S_{k}|$ is the number of points in that cluster.5.**Localization Refinement:** The astronaut’s final position is estimated as the cluster’s centroid that contains the most sensor data points within a given time window.To map the astronaut’s position into a global coordinate system, the position vector of the astronaut $p_{i}$ in the local coordinate frame of the sensor network is transformed into a global coordinate system $p_{i}^{\mathrm{global}}$ by applying a rotation and translation matrix. The transformation is given by(4)$$p_{i}^{\mathrm{global}}=Rp_{i}+T$$
where
$p_{i}^{\mathrm{global}}=\left(x_{i}^{\mathrm{global}},y_{i}^{\mathrm{global}},z_{i}^{\mathrm{global}}\right)$ is the astronaut’s position in the global coordinate system;R is the rotation matrix that accounts for the orientation difference between the local sensor frame and the global frame;T is the translation vector representing the offset between the origin of the local and global coordinate systems.

#### 3.4.3. IMU-Motion Capture

Motion capture recordings were performed separately for each traverse and task segment of the experimental procedure, resulting in recorded segments approximately 30 to 50 m in length. This approach was chosen to align with the experimental design and manage the motion capture system’s data volume, excluding periods of communication and rest. Data was processed using MVN Analyze Pro 2024.0. An extended Kalman filter (XKF3) was applied to fuse accelerometer, gyroscope, and magnetometer data from each sensor with a biomechanical model [40]. This model incorporates joint constraints and segment lengths to estimate the 3D orientation and position of each body segment in real time. The full-body kinematics are computed by solving the inverse kinematics of the model, constrained by the fused IMU data. The CoM trajectory is then derived from the body segment positions and masses. CoM trajectories were exported in the local coordinate system, linearly interpolated to 1 Hz, and synchronized with Aouda GPS data based on hardware timestamps. The position and orientation of each recording segment were transformed from the local coordinate frame to the environmental model coordinate system using initial GPS position and orientation at the start of each segment.

### 3.5. Statistical Analysis

The Median Euclidean Error (MEE) was calculated for each alternative localization method using GPS as the ground truth. This involved computing the 2D Euclidean distance at each time point and taking the median value. Under circular symmetric error assumption, the reported MEE would approximate the CEP, which is defined on geometric probabiltiy [41]. MEE from the different methods were statistically compared using a one-way analysis of variance (ANOVA). In the event of a significant ANOVA result, Bonferroni-corrected post hoc tests were performed to identify pairwise differences. When assumptions of normality were not met, the non-parametric Kruskal–Wallis test together with the post hoc Dunn’s test were used. All statistical analyses were performed in Python 3.12 using the SciPy Stats package (v1.15.3).

## 4. Results

The methodological approach proved feasible and the telemetry data from both Aouda suits and the iROCS rover could successfully be integrated and combined with the environmental model to create a prototypical digital twin for the AMADEE-24 analog mission in Armenia. Reconstructed telemetry from both suits includes locomotion state, heart rate, and helmet-internal temperature corrected for humidity and CO_2_ concentration, providing insights into the astronauts’ physiological condition during the EVAs. The operator models are still simplified and lack internal dynamic models to resemble complex system behavior. However, in this study, we aimed at “external” coherence of environmental model and operator models, primarily. Hence, the digital twin successfully enabled the reconstruction and detailed analysis of all nine analyzed experiment runs conducted during the mission. Figure 3 presents the visual representation of the digital twin from a broad view highlighting three experiment runs with different traverse paths reconstructed from GPS localization of the operators. Two photographs captured during the EVA on 17 March 2024, at EVA time 144:24 (min:sec), and at EVA time 219:04 (min:sec), are matched with their corresponding digitally rendered scene generated by the digital twin. Operator localization in the renderings were based on GPS coordinates. Additionaly, the digital twin enables the adjustment of sun position and lighting conditions, facilitating the evaluation of visual perception and situational awareness under various scenarios.

Overall structural and basic visual features are well matched, between the photographic and rendered images. Topographic shapes are well represented in the environmental model of the digital twin. However, perceptual differences remain, particularly in texture fidelity, blur, coloration, and minor artifacts. Notable differences can especially be observed in the foreground terrain, where fine details are blurred in the rendering, as well as in the texture and coloration of rover and astronauts. A standard mesh model from Blender was selected for the space suits as no high-resolution 3D mesh model of the Aouda suits exists so far.

Nine FarSiDE and iROCS experiment runs, including the traverses to the experiment sites, were analyzed. One run navigated the crew to a point of interest approximately 220 m west of the habitat, leading to a total covered distance by the rover of 1137.8 m. The astronauts traversed a total of 1667.2 m (AA(S)) and 1536.8 m (AA(R)). The other experiment runs were in closer proximity to the habitat. The distance covered by the iROCS rover during these runs was on average 283.9 ± 78.0 m, and it took them 61.2 ± 18.2 min. The analog astronaut performing the sampling tasks walked 520.2 ± 151.5 m, and the analog astronaut controlling the rover covered 377.6 ± 129.4 m on average during those experiment runs.

The MEE for rover localization using the SLAM algorithm was 1.51 ± 0.86 m (Med = 1.50 m). Building upon the SLAM localization of the iROCS rover, the cluster detection of the Aouda suits by the rover showed an MEE of 6.31 ± 2.32 m (Med = 5.63 m). The MEE for the astronaut localization using the IMU-MoCap was 2.42 ± 1.38 m (Med = 2.13 m). The differences in MEE between the alternative localization methods were statistically significant (H(2) = 17.96, *p* < 0.001). Post hoc Dunn’s tests indicate that iROCS SLAM (*p* < 0.001) and IMU-based motion capture (*p* = 0.041) are more accurate than Aouda Cluster Detection (Figure 4). No significant differences were found between the former two localization methods (*p* = 0.239). However, the differences in synchronization and calibration with the GPS position between the methods must be considered when interpreting these findings. The IMU-based motion capture path was calibrated independently for each recorded segment. At the same time, the cluster detection by the iROCS rover was continuous with a single GPS calibration at the start of the experiment run. One experiment run showed massive outliers in the motion capture localization (MEE = 5.89 m) due to signal transmission issues of the IMU sensors, leading to strong drift in the position data of the sensor-integrated Xsens model (Appendix A: Figure A1). Rapid turning of the rover around its axis led to difficulties in the SLAM algorithm, resulting in a high positioning offset error in two experiment runs (Appendix A: Figure A6 and Figure A7). One could be corrected by recalibrating with GPS position. As the Aouda cluster detection positioning is based on the precise positioning of the rover, error propagation also led to outliers in the MEE of the astronaut positioning. The astronaut leaving the field of view of the rover’s LiDAR sensor or another object entering the line of sight can lead to gaps in the position data, or, in the case of incorrect cluster detection, to deviations from the astronaut’s actual position. An exemplary localization comparison for the EVA on 27 March 2024 is shown in Figure 5. The GPS position, which was defined as ground truth for the comparisons, is shown as a colored traverse to indicate time progression over the course of the experiment. iROCS SLAM localization showed low deviation to GPS position in this run (MEE = 1.14 m). Additionally, the IMU-MoCap localization of the astronaut performed well (MEE = 1.23 m).

## 5. Discussion

### 5.1. Alternative Local Positioning Approaches

The results demonstrate that local positioning approaches can offer competitive accuracy for analog mission navigation without relying on GNSS. The outcomes are especially notable given that the commercial GPS module in the Aouda suits used as a benchmark has a reported CEP of up to 2.5 m—indicating that both the SLAM-based rover localization and IMU-MoCap tracking of astronauts can outperform or match GPS in this context. The findings align with previous research reporting localization errors of similar magnitude for IMU-based approaches, though differences in methodology and testing environments limit direct comparability. Reported errors range from approximately 1.2 m for short, simplistic EVA paths [9] to between 2 and 8 m for more complex, EVA-like traverses [10], reflecting the variability observed in the literature. However, a key limitation of this approach remains the cumulative translational drift in the absence of a global reference system, which becomes more pronounced over longer distances. In our study, continuous IMU-based motion capture over the whole duration of the experiment was not feasible due to data volume constraints. Therefore, the reported accuracy in our results must be interpreted with caution, primarily because of the periodic recalibration using GPS data. These external corrections would not be available in a GNSS-denied environment. In our study, data transmission issues between the individual IMUs and the receiver introduced sensor integration errors and artifacts in CoM trajectory, which contributed most to localization deviations (Appendix A: Figure A2, Figure A5, and Figure A6). A potential solution would be an integrated, wired IMU network within the space suit, leveraging Aouda telemetry for data transfer instead of individual wireless antennas. Future work should focus on advanced compression of motion capture data and continuous alternative recalibration mechanisms for position and orientation in GNSS-denied environments to support scalable inertial localization in analog and future lunar surface missions.

LeGO-LOAM-BOR SLAM demonstrated reliable performance in generating consistent maps and reliable localization over extended trajectories despite not integrating GNSS. However, limitations were observed during rapid rotations of the rover, which led to temporary localization drift and degraded map quality. To address performance degradation during rapid rotations, we analyzed contributing factors and identified two primary causes: the limited horizontal resolution of the LiDAR sensor and the inherent sensitivity of feature-based SLAM algorithms like LeGO-LOAM-BOR to motion distortion. First, LiDAR sensors with lower angular resolution (e.g., 0.2°–0.4° horizontal) inherently provide fewer points per scan when the platform undergoes fast yaw rotations. This results in sparse or unevenly distributed features in the angular domain, which in turn affects the robustness of scan registration and feature matching. During rapid rotations, the density of key edge and planar features drops significantly in regions perpendicular to the rotation axis, leading to less reliable pose estimation. Second, feature-based SLAM algorithms like LeGO-LOAM-BOR assume relatively smooth and continuous motion between scans. Rapid rotational movements violate this assumption, introducing motion distortion within a single scan due to the LiDAR’s sequential firing pattern. While LeGO-LOAM incorporates basic motion compensation using the rover’s IMU data, its effectiveness is limited at high angular velocities. This can lead to temporal misalignment between extracted features and reduce the accuracy of both the odometry front-end and the map-matching back-end. Future work could improve robustness under such conditions by increasing the LiDAR scan rate and angular resolution. Alternatively, more advanced motion compensation techniques—such as tightly coupled LiDAR–IMU fusion using continuous-time trajectory representations—could be integrated.

The Adaptive Clustering algorithm successfully detected astronauts in LiDAR point cloud data, but its performance was limited under certain conditions. One key issue was error propagation: spatial inaccuracies introduced during iROCS SLAM localization —upon which the Cluster Detection was calculated— compounded the astronauts’ positional uncertainty (Appendix A: Figure A7), reducing Cluster Detection localization accuracy. Additionally, the sparse point cloud caused by the LiDAR’s limited vertical resolution led to irregular cluster shapes. This made it difficult to compute consistent centroids, resulting in higher variation around the astronaut’s true position. Finally, the algorithm required a clear line of sight between the rover and astronauts. As shown in Appendix A: Figure A4 (100–110 min), occlusion could cause false positive detections, with the system occasionally misidentifying other individuals as astronauts.

Overall, these findings highlight the potential of LeGO-LOAM for large-scale planetary exploration in unstructured environments, while also indicating the need for higher-resolution LiDAR systems and refined clustering strategies for more robust astronaut detection.

### 5.2. Advanced Mission Planning Using Digital Twins

Integrating diverse sensors and data sources for generating a digital twin of the AMADEE-24 analog Mars mission, following the framework proposed by Reimeir et al. [19], proved feasible. EVAs were successfully reconstructed, demonstrating the potential of this approach. Factors contributing to some noticeable visual differences between selected EVA scenes and their corresponding digital twin renderings included the resolution of aerial images and discrepancies in the 3D mesh of operator models. The digital twin offers several potential applications for future analog and planetary missions:Pre-mission planning: The digital twin enables the Remote Science Team and Flight Planning Team to virtually explore the terrain and pre-select POI and traverse paths. However, practical usability must be evaluated in future deployments.Situational awareness: The digital representation of terrain and surface conditions may enhance situational awareness for the MSC. This includes a clearer understanding of operator position, environmental hazards, and spatial relationships between POI.Post-EVA analysis: The system facilitates detailed EVA reconstruction, enabling the analysis of traverse paths, task durations, time allocation, HRI, and potentially hazardous situations. When combined with physiological telemetry, such analyses support refining operational procedures and time scheduling.Mission evaluation: The digital twin contributes to post-mission assessment by supporting the evaluation of critical performance metrics [42].

Some limitations currently affect the perceptual fidelity and functionality of the digital twin. The environmental model relies on 2D aerial imagery mapped onto complex terrain, which can introduce distortions in color and structure, particularly for geological formations and fine structural elements. Incorporating high-resolution rover imagery from multiple angles could substantially improve texture accuracy and visual realism. So far, the system is limited to post-EVA reconstructions due to constraints on telemetry transmission. In rough or hilly terrain, maintaining a stable network across the entire region of interest presents a significant challenge. Both localization alternatives explored in this study are considerably more data-intensive than GNSS-based localization, and current bandwidth limitations prevent reliable real-time data transfer. We are currently developing a real-time digital twin; however, the first implementation will rely solely on GNSS data due to these transmission constraints.

## 6. Conclusions

This study demonstrated the feasibility of a digital twin for analog Mars missions, highlighting its potential value in mission planning and post-EVA analysis. The proposed GNSS-independent LiDAR- and IMU-based approaches offer a promising alternative for localizing astronauts and rovers in GNSS-denied environments. Fusing both approaches could effectively tackle some limitations by utilizing LiDAR information for IMU drift compensation and relying on the robustness of IMU-MoCap when a line of sight is obstructed or rapid rotations occur. Ongoing developments aim to enhance the digital twin to provide real-time integration of telemetry data and increase visual realism through high-resolution, ground-based imaging—both essential for operational support and scientific interpretation. Rover capabilities are being expanded to enable dynamic synchronization with astronaut activities via stereo vision and a real-time human tracking system, allowing the rover to adjust its behavior in response to astronaut posture and movement. Together, these developments strengthen the digital–physical interface of analog mission infrastructure, supporting the long-term goal of enabling safe, efficient, and scientifically productive planetary exploration.

## Figures and Tables

**Figure 1 sensors-25-04615-f001:**
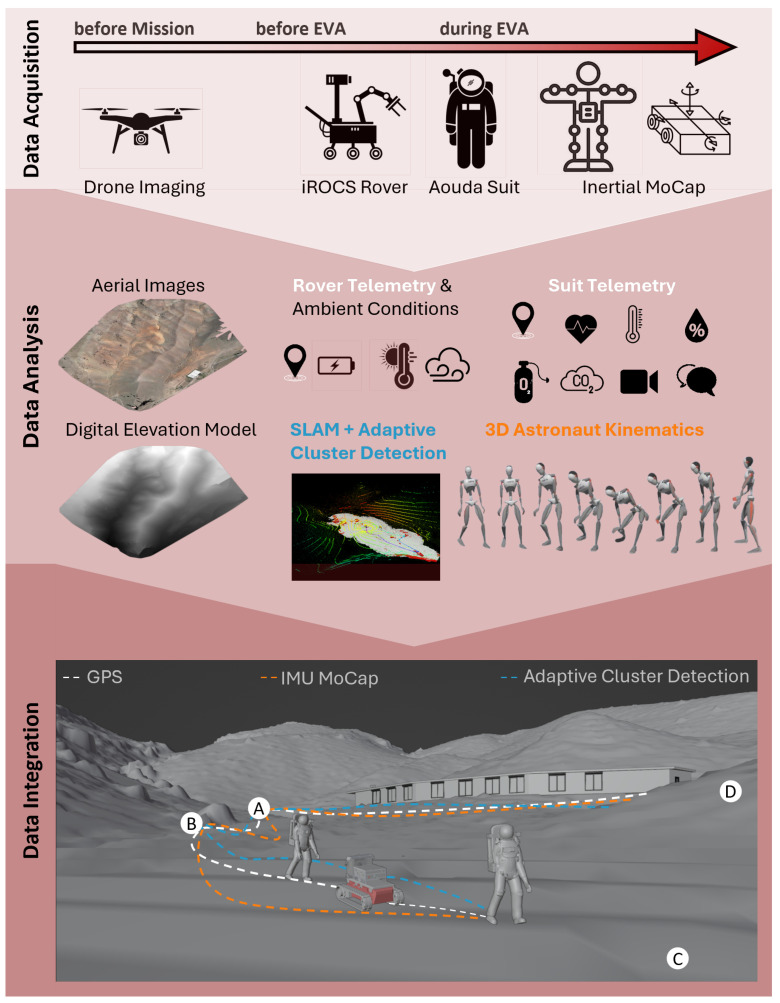
Study design and workflow of the digital twin integration. The figure illustrates the chronological deployment of devices and sensors from left to right. Vertically, the workflow progresses from data acquisition, through preprocessing and analysis, to the final integration and synthesis within the digital twin system. The bottom image presents a visual representation of the digital twin, incorporating the proposed experimental design for evaluating alternative localization methods based on the different integrated data. Points (A–D) represent the four points of interest visited during the EVA.

**Figure 2 sensors-25-04615-f002:**
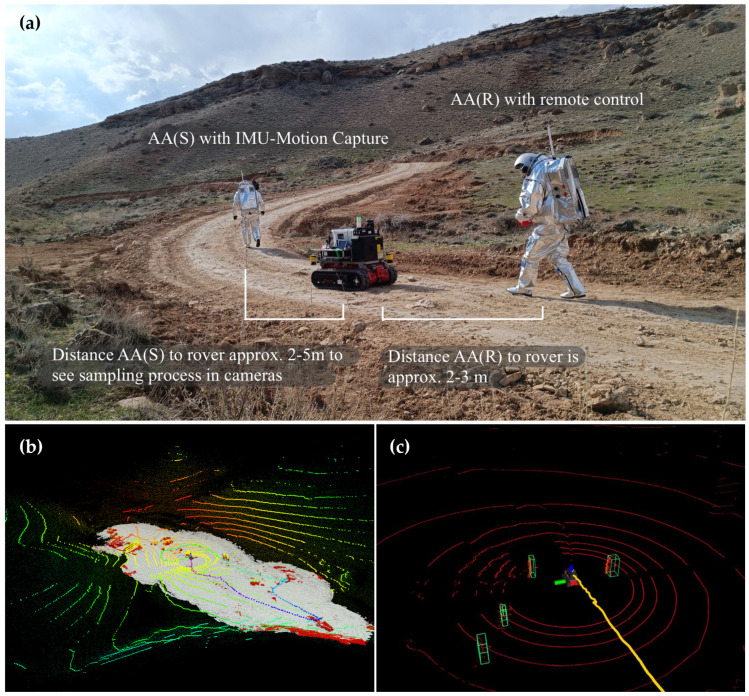
(**a**) Experimental design and instructions for testing alternative localization methods. (**b**) Universal Multi-Layer Map (UMM): LiDAR scans of the surroundings with point cloud generation. The traversability map is in grey, and the terrain hazards are in red. The rover localization is shown as a color sequence. (**c**) Adaptive Cluster Detection and localization of the astronauts. Two astronauts close to the rover were detected. Two additional individuals were detected further away. The path of the rover is shown in yellow.

**Figure 3 sensors-25-04615-f003:**
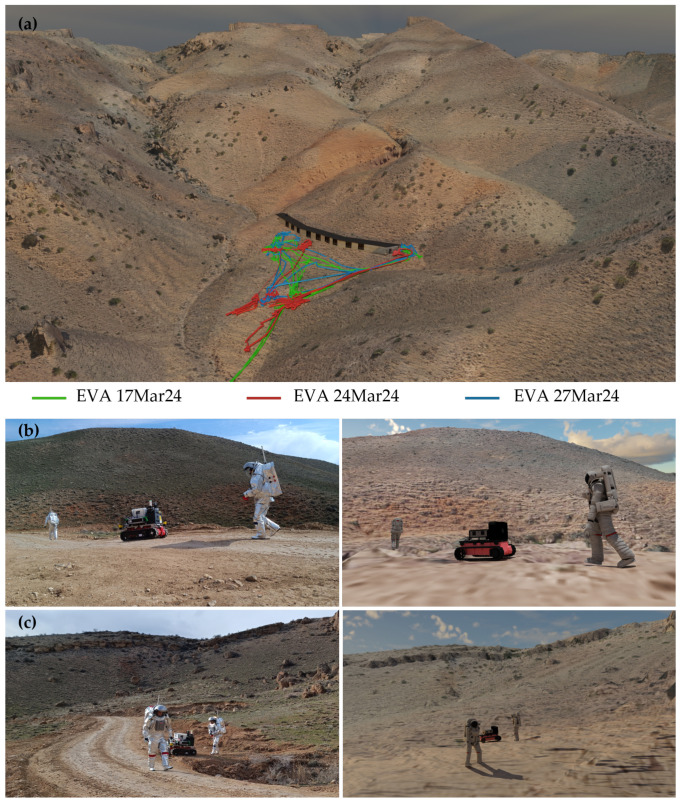
(**a**) Digital twin representation of the habitat region, where the experiment runs were conducted. Three exemplary traverses are highlighted. Photographic images of the two astronauts and the rover traversing, taken during the EVA on 17 March 2024 at EVA time (**b**) 144:24 (min:sec) and (**c**) 219:04 (min:sec) and their corresponding rendered scenes from the digital twin.

**Figure 4 sensors-25-04615-f004:**
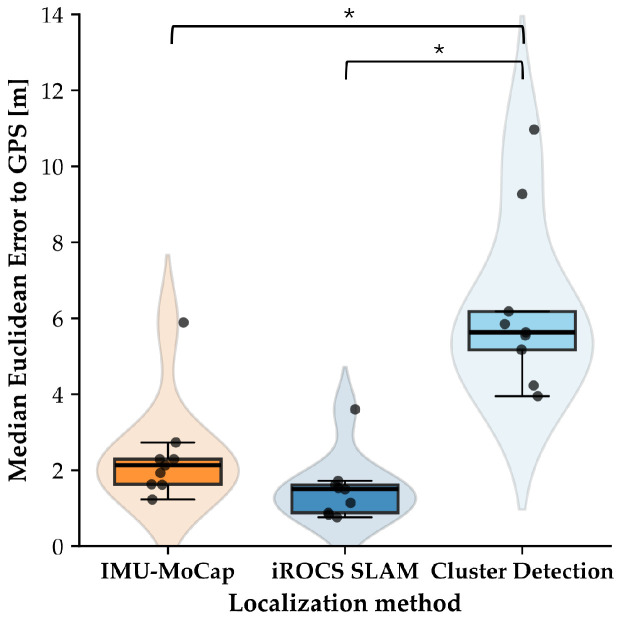
MEE between the three proposed localization methods and conventional GPS positioning. MEE was calculated for each experiment run. IMU-MoCap-based localization was conducted segment-wise rather than continuously, requiring caution in direct comparison and interpretation. Significant differences are marked with *. n = 9.

**Figure 5 sensors-25-04615-f005:**
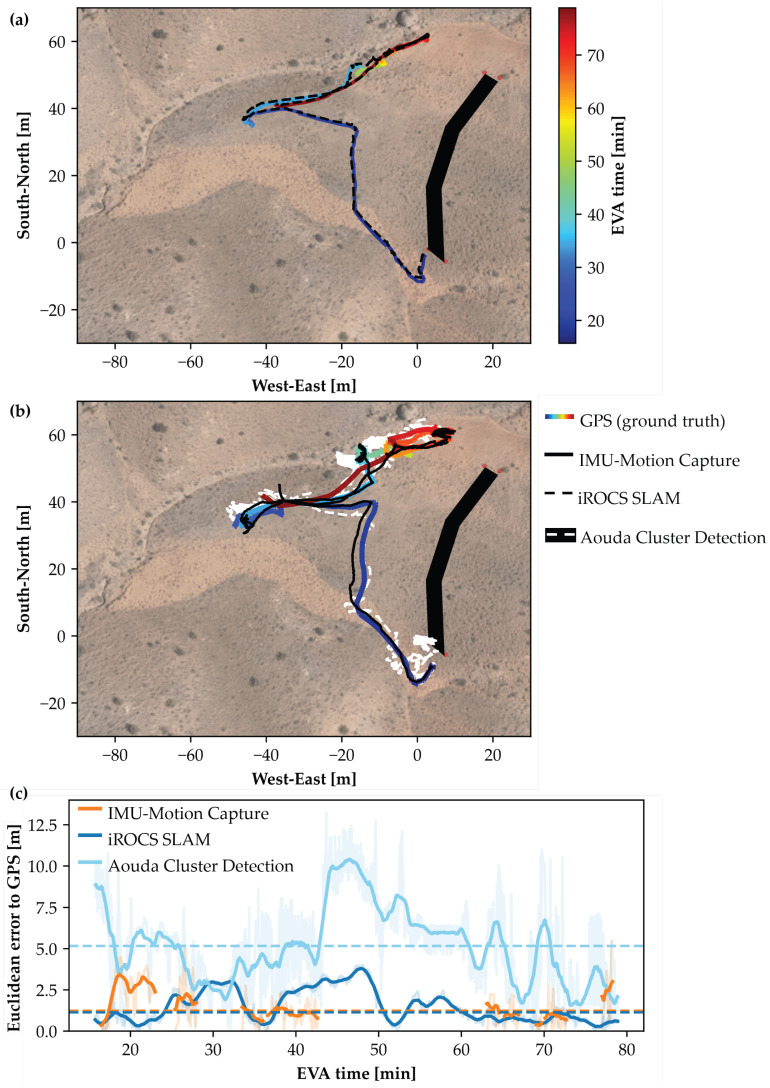
(**a**) Rover and (**b**) astronaut positions of the first experiment run during the EVA on 27 March 2024. GPS traverse paths, serving as ground truth, are compared to SLAM localization of the iROCS rover and IMU-MoCap-based localization and Cluster Detection of the astronaut. Habitat is depicted as the black complex in the east. (**c**) Euclidean error between the GPS position and the different methods over time. IMU-MoCap localization is intermittent due to the recording protocol. Large astronaut–rover distance led to Aouda cluster misdetection, resulting in increased errors.

## Data Availability

The Aouda suit telemetry data is publicly available via the OeWF Science Data Archive. IMU motion capture data and iROCS rover data are available upon request.

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
