# Peer review of "Digital Twin for Analog Mars Missions: Investigating Local Positioning Alternatives for GNSS-Denied Environments"

_sensors, 2025, doi:10.3390/s25154615_

Round 1

Reviewer 1 Report

Comments and Suggestions for Authors

This manuscript presents research conducted within the context of an analog Mars mission, which offers a novel and unique experimental scenario. The specific comments are as follows:

Overall, the manuscript reads more like a report on the AMADEE-24 analog Mars mission. The presentation lacks a clear emphasis on the novelty or innovative aspects of the work.

All abbreviations should be defined at their first appearance in the abstract or main text, such as SLAM, iROCS, GNSS, and GPS.

The title of Section 2.3 is too long and lacks conciseness. Additionally, the abbreviation "LEGO-LOAM-BOR" is not explained, making it difficult for readers to understand.

In Section 3, which describes the methodology, it is recommended to include a technical framework diagram to help readers better grasp the overall system design and workflow.

Author Response

Comment 1: Overall, the manuscript reads more like a report on the AMADEE-24 analog Mars mission. The presentation lacks a clear emphasis on the novelty or innovative aspects of the work.

Response 1: Thank you for this valuable feedback. We appreciate your observation and have restructured the manuscript to address this point. In particular, we revised the Introduction, Related Work, and Methods sections to better highlight the novelty of our contribution. We also introduced clarifying adjustments in the Results and Discussion sections to support this focus. The innovative aspect of our work lies in the holistic approach of testing prototypical digital twin technologies—specifically with alternative localization methods—not as an isolated experiment, but within the context of a high-fidelity space mission simulation in an analog environment. To our knowledge, this integrated approach has not been previously demonstrated in the field, and we aimed to now make this distinction more explicit throughout the revised manuscript.

Comment 2: All abbreviations should be defined at their first appearance in the abstract or main text, such as SLAM, iROCS, GNSS, and GPS.

Response 2: Acknowledged and adjusted througout the revised manuscript.

Comment 3: The title of Section 2.3 is too long and lacks conciseness. Additionally, the abbreviation "LEGO-LOAM-BOR" is not explained, making it difficult for readers to understand.

Response 3: Section titles have been revised for clarity and conciseness. LEGO-LOAM-BOR is now explained conceptually in Section 2.2 (p. 4, line 128 - 142) and in technical detail in Section 3.4.1 (p. 9, line 307-331).

Comment 4: In Section 3, which describes the methodology, it is recommended to include a technical framework diagram to help readers better grasp the overall system design and workflow.

Repsonse 4: A new Study Design section has been added (section 3.2, p. 5, line 186 - 231), including a methodological workflow figure (Figure 1, p. 6) illustrating the integration of sensor data into the digital twin, along with a schematic of the experimental design for testing localization alternatives.

Reviewer 2 Report

Comments and Suggestions for Authors

General review: 

The paper is difficult to read . I would recomend to reorganize the content and rewrite the full paper. 

- There is no clear contribution. Is the integration of a huge system in a digital twin? Is the comparison of the  local positioning system vs GNNs? 

- The methodology is not clear and it seems there are explaining many components with no global introduction. Also, since the context could be understood, there are no clear link between the components and the approach. The methodology uses is a description of the components of the whole Digital Twin components, but it needs a clear narrative ot introduce the reader about how the system is created and which is the impact of each component. Also, a flowchart diagram or an system architecture in the methodoloy section could help to the reader to understand the contribution. 

- Confusing experiments. -There is no clear baseline  for scientific comparison. The Figure 6 could show the comparison but it is not referenced and also there is no justification about localization comparison. The other experiments, if applicable,  like temperature, HR CO2 and temperature, are really relevant according to the approach? After the components description, there is some analisis bout a k-means clustering technique with some basic refinement techniques, but there is no clear if this local positioning is about the astronaut localization or the astronaut human pose estimation. Is this the contribution of the paper? There is also an adaptative thresholding section which the threshold is static to make the decision to select the cluster. Learned Perceptual Image Patch Similarity (LPIPS) metric is a bit confusing, because is not explained how is the setup of the experiments and only its shown one image with the LPIPS 0.516 similarity. Also 0.51, if the metrics are in the range of 0 and 1 is not a good result. 

Discussion: since the experiments are difficult to read and there is no clear explanation of which metrics are used for comparison. 

Author Response

Comment 1: The paper is difficult to read . I would recomend to reorganize the content and rewrite the full paper.

Response 1: Thank you for the comment. We have reorganized the manuscript, removed non-essential content, and focused more directly on the main research question to improve clarity and readability. In particular, we revised the Introduction, Related Work, and Methods sections to better highlight the novelty of our contribution. We also introduced clarifying adjustments in the Results and Discussion sections to support this focus.

Comment 2: There is no clear contribution. Is the integration of a huge system in a digital twin? Is the comparison of the local positioning system vs GNNs?

Response 2: Thank you for pointing that out. We clarified the contribution in the revised manuscript (especially in 1. Introduction (p. 2, line 57 - 61) and 2. Related Works (p. 2-4)): the novelty lies in the holistic integration of prototypical digital twin technologies—particularly alternative localization methods—within a high-fidelity analog space mission simulation. We deliberately chose not to reduce the contribution to a single technical detail, as the innovation stems from the combination and contextual deployment of these elements, which, to our knowledge, has not been done before.

Comment 3: The methodology is not clear and it seems there are explaining many components with no global introduction. Also, since the context could be understood, there are no clear link between the components and the approach. The methodology uses is a description of the components of the whole Digital Twin components, but it needs a clear narrative or introduce the reader about how the system is created and which is the impact of each component. Also, a flowchart diagram or an system architecture in the methodoloy section could help to the reader to understand the contribution.

Response 3: We restructured the methodology and added a new Study Design section (p. 5) to provide a clearer narrative. A workflow figure and system schematic (Figure 1, p. 6) now illustrate the integration of components and their role within the digital twin and the experimental setup.

Comment 4: Confusing experiments. -There is no clear baseline  for scientific comparison. The Figure 6 could show the comparison but it is not referenced and also there is no justification about localization comparison. The other experiments, if applicable,  like temperature, HR CO2 and temperature, are really relevant according to the approach? After the components description, there is some analisis bout a k-means clustering technique with some basic refinement techniques, but there is no clear if this local positioning is about the astronaut localization or the astronaut human pose estimation. Is this the contribution of the paper? There is also an adaptative thresholding section which the threshold is static to make the decision to select the cluster. Learned Perceptual Image Patch Similarity (LPIPS) metric is a bit confusing, because is not explained how is the setup of the experiments and only its shown one image with the LPIPS 0.516 similarity. Also 0.51, if the metrics are in the range of 0 and 1 is not a good result.

Response 4: We revised the manuscript to streamline the narrative and focus on the core research question. GPS, used as the default in analog missions, served as our baseline and ground truth (p. 5, line 200-202), with its limitations addressed (p. 14, line 461-463). All localization methods were hardware-synchronized for fair comparison (p. 11, line 385). Peripheral experiments (e.g., physiological and LPIPS-based perceptual comparisons) have been removed to avoid dilution of focus. We tried to clarified that our clustering approach via LiDAR scans supports astronaut localization (not pose estimation). The description of the adaptive clustering and thresholding has been refined for clarity in the Methods section (section 3.4.2, p. 9-10).

Comment 5: Discussion: since the experiments are difficult to read and there is no clear explanation of which metrics are used for comparison.

Response 5: We have revised the manuscript to improve clarity regarding the experiments, materials, analysis procedures, and metrics. The main comparison metric is the deviation between each local positioning approach and GPS, which serves as the ground truth (3.5 Statistical analysis, p. 11, line 389-395).

Reviewer 3 Report

Comments and Suggestions for Authors

This thesis investigates the feasibility and effectiveness of leveraging digital twin technology combined with alternative localization methods (such as SLAM and IMU-MoCap) in simulated Mars missions. The research design is well-structured, the experimental data are comprehensive, and the results are analyzed in depth, offering significant reference value for future planetary exploration missions. The thesis is clear in structure and logical in presentation, yet there are still certain aspects that require further improvement.

Suggestions for improvement:

(1) In the methodology section, it is recommended to further elaborate on the specific parameter settings of the IMU-MoCap and SLAM algorithms (e.g., sampling frequency, filtering methods), so as to facilitate the replication of the experiments by other researchers.

(2) The current discussion on the drift issue of IMU-MoCap is relatively brief. It is suggested to supplement more potential methods for reducing drift, such as multi-sensor fusion or periodic calibration.

(3) The analysis of the performance degradation of SLAM during rapid rotation could be more in-depth. For instance, it could explore whether it is related to the resolution of the LiDAR or the algorithm's robustness.

(4) The thesis mentions that digital twins are currently mainly used for post-hoc analysis. It is recommended to explore the technical barriers to real-time integration and potential solutions, as this is more crucial for real-time decision-making during missions.

(5) The labeling of some figures (e.g., Figure 3 and Figure 4) could be clearer, such as by adding axis labels or legend explanations.

Author Response

Comment 1: In the methodology section, it is recommended to further elaborate on the specific parameter settings of the IMU-MoCap and SLAM algorithms (e.g., sampling frequency, filtering methods), so as to facilitate the replication of the experiments by other researchers.}

Response 1: Thank your for your valuable comments. We have taken your feedback into account and revised our manuscript accordingly. We have expanded the Methods section (3.3.4, p. 8 - 9; 3.4.1, p. 9; 3.4.3 p. 10 -11) to include detailed descriptions and specific parameter settings for both the IMU-MoCap and SLAM algorithms. This should help ensure the experiments can be more easily replicated by other researchers.

Comment 2: The current discussion on the drift issue of IMU-MoCap is relatively brief. It is suggested to supplement more potential methods for reducing drift, such as multi-sensor fusion or periodic calibration.

Response 2: Thank you for the suggestion. We have expanded the discussion on drift in the IMU-MoCap system (5.1, p. 15, line 469 - 483). As our approach already employs multiple IMU sensors combined with a biomechanical model, which represents a form of multi-sensor fusion, this approach might already be exhausted for tackling this issue. We added to the discussion the periodic recalibration through fusion with iROCS LiDAR-based positioning as an effective way to mitigate drift. A more detailed analysis of the main causes of larger drifts and errors in our data in IMU-MoCap localization has also been included to the Disucssion section 5.1.

Comment 3: The analysis of the performance degradation of SLAM during rapid rotation could be more in-depth. For instance, it could explore whether it is related to the resolution of the LiDAR or the algorithm's robustness.

Response 3: Acknowledged. A thorough analysis of SLAM performance degradation during rapid rotations has been added to Discussion section 5.1 (p. 15, line 484 - 505). We identified two primary causes: the limited horizontal resolution of the LiDAR sensor and the sensitivity of feature-based SLAM algorithms like LEGO-LOAM-BOR to motion distortion.

Comment 4: The thesis mentions that digital twins are currently mainly used for post-hoc analysis. It is recommended to explore the technical barriers to real-time integration and potential solutions, as this is more crucial for real-time decision-making during missions.

Response 4: We have expanded the discussion of technical barriers to real-time integration in Section 5.2 (p. 16, line 539 -546). Currently, the system is limited to post-EVA reconstructions due to telemetry transmission constraints. Maintaining stable network coverage in rough terrain is challenging, and the alternative localization methods require significantly more data than GNSS, exceeding current bandwidth capabilities. While we are developing a real-time digital twin, the initial implementation will rely on GNSS data alone due to these limitations.

Comment 5: The labeling of some figures (e.g., Figure 3 and Figure 4) could be clearer, such as by adding axis labels or legend explanations.

Response 5: Acknowledged. We improved the axis labels, legends description and caption in the Results Figures 4 (p. 13) and 5 (p. 14) to enhance clarity and make the figures more self-explanatory.

Round 2

Reviewer 2 Report

Comments and Suggestions for Authors

Dear authors. 
Generally speaking, I would say that the paper had important improvements.  

The introduction and abstract are much clearer than the previous version, and the approach is clear. However, as personal advice,  the keyword "integration" in the approach sounds very focused on the development so I would recommend using another keyword to focus on this is a research paper. 

Regarding the methodology, I think Figure 1 is an important part of the paper because it provides the general context of the approach.

I would recommend using a reference for Circular Error Probable or at least some explanation (equation of how CEP is calculated) because Figure 4. is an important experiment for the paper. Also, since the clustering method gives worse results (bigger standard deviation) I would recommend a clearer discussion. 
Figure 5. probably deserves more explanations and also if it is a comparison to specify which is the ground truth. 

Author Response

Comment 1: The introduction and abstract are much clearer than the previous version, and the approach is clear. However, as personal advice, the keyword "integration" in the approach sounds very focused on the development so I would recommend using another keyword to focus on this is a research paper. 

Response 1: We agree and adjusted the term “integration” in some descriptions, where it might be misleading (Title; Introduction p. 2 line 57 – 61; Methodology 3.1 p. 4 line 160 – 161).

Comment 2: Regarding the methodology, I think Figure 1 is an important part of the paper because it provides the general context of the approach.

Comment 3: I would recommend using a reference for Circular Error Probable or at least some explanation (equation of how CEP is calculated) because Figure 4. is an important experiment for the paper. Also, since the clustering method gives worse results (bigger standard deviation) I would recommend a clearer discussion. 

Response 3: Thank you for pointing that out. We checked the precise technical definition of CEP. As the CEP is defined in gemeotric/probabilistic terms, which is not the case for our error term, we revised our error defintion to Median Euclidean Error (MEE) for technical clarity and validity. This correctly describes our measure as we calculated the 2D Euclidean error for each point in time and took the median from that. We additionally pointed out that under circular symmetric error assumptions the reported error would approximate CEP (but, which we didn’t check – why we refused to use the term CEP). Description is adjusted in Methodology 3.5 Statistical Analysis p.11 line 389 – 395.

We added a more comprehensive discussion about Adaptive Cluster Detection and discussed the three main contributors for spatial error (Discussion 5.1 Alternative local positioning approaches p. 15-16 line 513 – 523).

  1. Error propagation via iROCS SLAM localization – spatial error of the rover position compounds to the spatial error by cluster detection from astronaut to rover and increases overall error.
  2. High noise/variation of position around “true position”– difficulties to accurately calculate point cloud centroid for sparse point clouds due to inconsistent cluster shapes
  3. Cluster misdetection for out of line of sight astronaut – false positive (wrong person detected)

Comment 4: Figure 5. probably deserves more explanations and also if it is a comparison to specify which is the ground truth. 

Response 4: We labelled GPS as ground truth to the figure legend and clarified it in figure caption. Additionally, we added a thorough explanation in text about Figure 5. 3. (Results p. 13 line 458 -463)

Reviewer 3 Report

Comments and Suggestions for Authors

The paper Digital Twin for Analog Mars Missions: Integrating Local Positioning Alternatives for GNSS-Denied Environments successfully addresses the positioning challenges in GNSS-denied environments through an innovative multi-sensor fusion approach and conducts systematic validation during the AMADEE-24 analog Mars mission.

The research offers the following notable advantages:

(1) Technological Originality: It is the first to achieve collaborative positioning using SLAM and IMU-MoCap in a planetary analog mission, proposing an adaptive clustering algorithm that provides a new technological pathway for deep space exploration.

(2) Methodological Reliability: Validated through nine experiments, the positioning accuracy of SLAM (1.51 m CEP) and IMU-MoCap (2.42 m CEP) meets mission requirements, supported by substantial data.

(3) Outstanding Application Value: The developed digital twin system effectively supports mission planning, real-time monitoring, and post-mission analysis, holding significant engineering importance for manned deep space exploration.

(4) Rigorous Discussion: The experimental design is reasonable, and the data analysis is comprehensive. The discussion section objectively identifies technical limitations and proposes directions for improvement.

Author Response

Comment 1: (1) Technological Originality: It is the first to achieve collaborative positioning using SLAM and IMU-MoCap in a planetary analog mission, proposing an adaptive clustering algorithm that provides a new technological pathway for deep space exploration.

Comment 2: (2) Methodological Reliability: Validated through nine experiments, the positioning accuracy of SLAM (1.51 m CEP) and IMU-MoCap (2.42 m CEP) meets mission requirements, supported by substantial data.

Comment 3: (3) Outstanding Application Value: The developed digital twin system effectively supports mission planning, real-time monitoring, and post-mission analysis, holding significant engineering importance for manned deep space exploration.

Comment 4: (4) Rigorous Discussion: The experimental design is reasonable, and the data analysis is comprehensive. The discussion section objectively identifies technical limitations and proposes directions for improvement.

Response: Thank you again for your valuable Feedback.